# ♣♣R²AG: Learning to Reason, Retrieve, and Self-Evolve through a Multi-Branch Retrieval Tree

## Abstract

Retrieval-Augmented Generation (RAG) boosts large language models (LLMs) with external retrieval. Despite its effectiveness, RAG struggles on complex multi-hop tasks due to the retriever's inability to recover sufficient evidence chains based on previously retrieved context in single-step search. To mitigate this, we adopt a *step-wise retrieval* paradigm that retrieves evidence chains through successive retrievals conditioned on previously retrieved context. We conceptualize this process as constructing a multi-branch retrieval tree, rooted at the original query, where each branch represents a context-dependent retrieval chain. This reformulation motivates a *reasoning-driven strategy* for tree expansion. Thus, we propose *Reasoning-Augmented RAG (R²AG)*, which *post-trains a retrieval steering model via reinforcement learning (RL) to drive the reasoning-augmented expansion of a multi-hop retrieval tree* (i.e., Reasoning-Augmented Retrieval Tree, RRT), grounded in multi-hop context comprehension. Yet a new challenge arises: as branches explode and distracting retrievals accumulate, expanding along precise chains in subsequent steps becomes intractable. To address this, we propose *Top-Survivor*, a method that *selects the accurate branches in RRT for further expansion under strict gold-hit based constraints*. Moreover, R²AG adopts a *progressively iterative* training design, enabling the model to incrementally *self-evolve* its reasoning capability by constructing and refining RRT through learning from accurate chains. Extensive experiments demonstrate that our method substantially improves the retrieval quality of RAG. Compared to naive RAG, R²AG yields 24.1%/20.4% gains in recall/accuracy, without incurring significant latency.

## 1 Introduction

Despite the breakthrough improvements in the performance of LLM, the issues of hallucination and outdated knowledge remain persistent challenges for LLM (Xu et al., 2024a). The RAG method (Lewis et al., 2020; Guu et al., 2020), by retrieving query-relevant documents from external knowledge bases as generation context, effectively addresses these issues and substantially enhances LLM performance in knowledge-intensive tasks (Ram et al., 2023).

RAG relies on a retriever to retrieve $N$ documents as context for answering a query. The effectiveness of RAG heavily depends on *retrieval quality*, i.e., the retriever's ability to surface sufficient gold evidence (Gao et al., 2023). While RAG performs well on simpler tasks such as single-hop QA, where a single relevant document suffices, it struggles on complex settings such as multi-hop tasks. These tasks require aggregating multiple pieces of evidence with implicit sequential multi-hop dependencies, which are typically *inaccessible via single-step retrieval* (Li & Peng, 2023).

Existing methods introduce step-wise retrieval, a *successive retrieval* paradigm where each step is conditioned on the previously retrieved content to recover multi-hop evidence (Huang & Huang, 2024). IRCoT (Trivedi et al., 2022b) adopts dynamically generated Chains-of-Thought (CoT) for step-wise retrieval, but may lead to error propagation and performance instability. Search-R1 (Jin et al., 2025) enhances both retrieval and generation via RL, while its outcome-based rewards may restrict retrieval strategy learning. HippoRAG 2 (Gutiérrez et al., 2025) performs graph-based multi-hop search yet suffers from semantic loss during corpus-to-triple conversion.

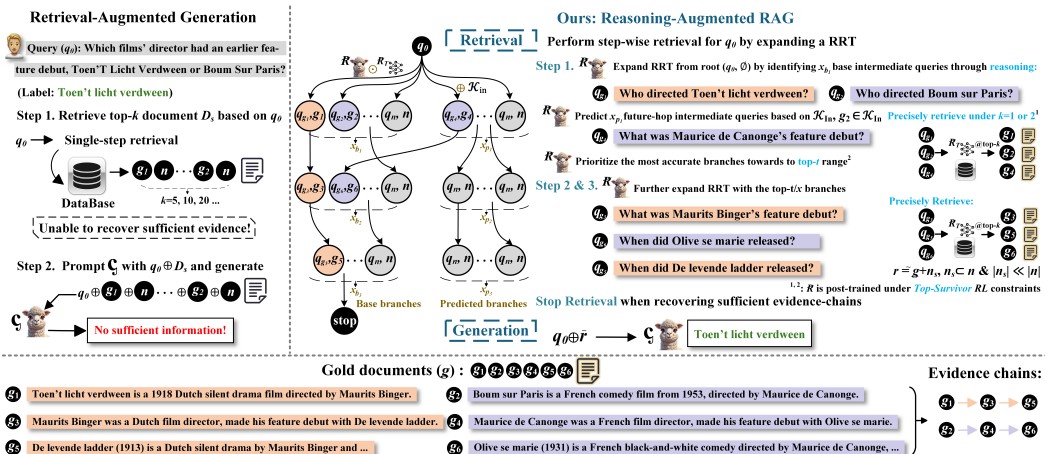

Figure 1: **Overview of R²AG.** R²AG enhances retrieval by performing step-wise retrieval. It frames this process as the construction and expansion of a Reasoning-Augmented Retrieval Tree (RRT), driven by a RL-trained retrieval steering model $\mathfrak{R}$. $\mathfrak{R}$ learns to effectively expand RRT by generating precise intermediate queries (boosting recall) and promoting accurate branches into top-$t$ candidates (boosting AP) under *Top-Survivor* constraints. $\mathfrak{R}$ also learns to make anticipatory predictions using internal knowledge $\mathcal{K}_{in}$ to accelerate retrieval. $\mathcal{G}$: generator model, $\mathfrak{R}_T$: retriever, $n$: noisy retrievals.

To mitigate the limitations of single-step retrieval, we adopt a step-wise retrieval paradigm and conceptualize it as constructing a multi-branch retrieval tree. Rooted at the original query, each vertex contains an intermediate query-retrieval pair, each edge denotes a retrieval operation by a retriever, and *each branch forms a context-dependent retrieval chain*. This reformulation naturally motivates our *Reasoning-Augmented RAG (R²AG)*—a *reasoning-driven* strategy for tree expansion. R²AG *post-trains a retrieval steering model* via RL to drive *the reasoning-augmented expansion* of a multi-branch retrieval tree (i.e., *Reasoning-Augmented Retrieval Tree, RRT*). Specifically, the model learns to generate a set of candidate intermediate queries for retrieval at each step, grounded in *multi-hop reasoning over the query and previously retrieved context*. This enables dynamic expansion of RRT in both breadth and depth, facilitating more sufficient evidence chains. Moreover, R²AG adopts *a progressive iterative training workflow*, where diverse previous retrievals serve as context for new RRT samples to fully develop the model's retrieval steering capability across varied scenarios.

Nevertheless, new challenges arise. First, as the RRT explosively expands in scale, it accumulates *substantial irrelevant retrievals*, rendering it *intractable to expand along accurate branches*. To address this, we propose *Top-Survivor*, a set of constraints based on *gold-hit* and *precision* signals. It restricts the retriever to return only the *top-1* document for each intermediate query and assesses evidence coverage and chain completeness, encouraging the steering model to *generate precise intermediate queries* and recover sufficient evidence—thus *boosting recall*. Additionally, it incorporates the average precision (AP)-based reward within the top-$t$ branches in each expansion, guiding the model to *prioritize accurate branches towards the top-$t$ candidates*, thereby enhancing precision and facilitating selection of accurate branches for further expansion. Second, the step-wise retrieval incurs *additional latency*. To mitigate this, we propose *anticipatory prediction*, which encourages the model to proactively predict the future-hop retrieval queries using its internal knowledge.

Extensive experiments demonstrate that R²AG *markedly boosts retrieval quality*. During R²AG training, the steering model progressively enhances its reasoning capability, enabling effective expansion of RRT both in breadth and depth. Namely, it learns not only to *refine previously expanded branches* in simpler RRTs, but also to *generalize to new, accurate branches* in complex samples—a process we term *self-evolution*. Moreover, comprehensive evaluations validate the effectiveness of R²AG in enhancing both retrieval and QA performance. Across four datasets, it yields average gains of *24.1%/17.5%* in *recall/mAP* over single-step retrieval methods, along with *20.4%* QA *accuracy* gains on six datasets—without incurring notable latency. Our contributions are as follows:

- We propose Reasoning-Augmented Retrieval Tree (RRT), which formalizes a reasoning-driven step-wise retrieval paradigm via an RL-post-trained retrieval steering model, and improves retrieval quality through effective and accurate reasoning-based expansion of RRT.

- We propose R$^2$AG, a framework that post-trains a retrieval steering model via RL to effectively expand RRT through reasoning, thereby enhancing retrieval quality. It introduces Top-Survivor constraints to guide the model in generating precise intermediate queries for higher recall and promoting accurate branches into top-$t$ candidates to improve precision.

- Extensive experiments demonstrate that R$^2$AG markedly boosts retrieval and QA performance by precisely retrieving sufficient evidence chains. Compared to single-step retrieval, R$^2$AG yields 24.1%/20.4% gains in recall/accuracy without incurring notable latency.

## 2 R$^2$AG: Learning to Reason, Retrieve, and Self-Evolve

Step-wise retrieval typically involves using a model to steer a multi-hop retrieval process by iteratively generating intermediate queries, with each step grounded in multi-hop comprehension of the query and previously retrieved documents. This process can be formalized as the construction of a multi-branch retrieval tree (Li et al., 2024). R$^2$AG further augments tree expansion through reasoning, resulting in what we term Reasoning-Augmented Retrieval Tree (RRT). An RRT is detailed as:

**Definition 2.1.** An RRT $= (V, E)$ is a reasoning-augmented retrieval tree rooted at the original query $q_0$. Each vertex contains a query-retrieval pair $v_{ij} = \{q_{ij}, \tilde{r}_{ij}\} \in V$, where $q_{ij}$ denotes an intermediate query generated by a retrieval steering model $\mathfrak{R}$ through *reasoning*, and $\tilde{r}_{ij}$ the associated retrieved documents. Each edge $e \in E$ denotes a retrieval operation by a retriever $\mathfrak{R}_T$. Let $h_i$, $h$, $b$ denote the depth of $i$-th branch, the maximum retrieval depth, and the number of branches in RRT.

Namely, each branch $B_i = \cup_{j=1}^{h_i}(v_{ij}, e_{ij})$ forms a context-dependent retrieval chain. The full set of retrieved documents is denoted as: $D_s = \cup_{i=1}^{b} \cup_{j=1}^{h_i} \tilde{r}_{ij} = \cup_{i=1}^{b} \cup_{j=1}^{h_i} \{g_{ij}, n_{ij}\}$, where $g$ and $n$ represent gold and noisy documents. *The central goal of R$^2$AG is to effectively expand the RRT to retrieve sufficient gold evidence within $D_s$,* formulated as ($G_T$ is the complete set of gold documents of $q_0$):

$$\epsilon = |G_T| - |G_s \cap G_T| \to 0, \ G_s = \cup_{i=1}^{b} \cup_{j=1}^{h_i} g_{ij} \leftarrow \text{RRT}. \tag{1}$$

### 2.1 Why Augmenting Retrieval Tree Expansion through Reasoning

The RAG system typically relies on either lexical or *dense vector similarity* to retrieve semantically relevant documents (Gao et al., 2023). The latter, which leverages dense embedding models, generally demonstrates superior performance in QA tasks (Abdallah et al., 2025).

**Assumption 2.1.** The retrievers discussed in this work typically rely on vector similarity to retrieve top-$k$ relevant documents. Moreover, the database does not contain semantically duplicate documents, which ensures that *retrievers avoid retrieving duplicate content in a single-step retrieval.*

The expansion of an individual branch $B_i$—that is, the expansion of a vertex $v_{ij} = \{q_{ij}, \tilde{r}_{ij}\}$ within $B_i$—involves generating the intermediate query $q_{ij}$ through reasoning over the original query $q_0$ and the cumulative retrieved documents $\cup_{u=1}^{j-1} \tilde{r}_{iu}$, following: $q_{ij} = \nabla_r(q_0 \oplus \sum_{1 \le u < j}^{\oplus} \tilde{r}_{iu})$, where $\nabla_r$ denotes a *reasoning operator* implemented by $\mathfrak{R}$ based on *Transformer's attention* mechanism; "$\oplus$" indicates the concatenation of different documents in natural language order. The retrievals $\tilde{r}_{ij}$ are obtained by the retriever $\mathfrak{R}_T$ based on vector similarity, selecting the top-$k$ most relevant documents. Thus, we have: $\tilde{r}_{ij} = \nabla_v(q_{ij}, D_B)$, where $\nabla_v$ denotes a *retrieval operator* implemented by $\mathfrak{R}_T$ that performs semantic relevance matching based on vector similarity, $D_B$ the database. Furthermore, we formalize the probability of retrieving a new gold document during the expansion step as:

$$p(\mathbf{E}_A) = \Phi\left[\pi_r(\nabla_r), q_0; \ \pi_v(\nabla_v), D_B\right], \quad \frac{\partial \Phi}{\partial \pi_r} \ge 0, \frac{\partial \Phi}{\partial \pi_v} \ge 0 \text{ and } \frac{\partial^2 \Phi}{\partial \pi_r^2} \neq 0, \frac{\partial^2 \Phi}{\partial \pi_v^2} \neq 0 \tag{2}$$

where $\mathbf{E}_A$ denotes event of retrieving a new gold document: $\exists d_u \in \tilde{r}_{ij} \Rightarrow d_u \in G_T, d_u \notin G_r$; here $G_r$ is the set of gold documents that are previously retrieved; $\pi_r$ denotes a differentiable policy function positively correlated with $\nabla_r$, and $\pi_v$ denotes one associated with $\nabla_v$ under fixed retrieval settings.

**Remark 2.1.** Since $q_0$ and $D_B$ are fixed, increasing the likelihood of retrieving new gold documents relies on either enhancing the semantic retrieval capability of $\nabla_v$ (i.e., $\mathfrak{R}_T$) (Thakur et al., 2021), or refining the reasoning-driven steering behavior of $\nabla_r$ (i.e., $\mathfrak{R}$) under fixed retrieval settings. It proves that by post-training the reasoning capability of $\mathfrak{R}$, the probability of retrieving gold documents during RRT expansion is increased, which facilitates more effective subsequent expansion.

## 2.2 TOP-SURVIVOR: DRIVING PRECISE RETRIEVAL AND ACCURATE BRANCH SELECTION

For a multi-hop query $q_0$, $\mathfrak{R}$ is typically required to progressively identify a sequence of single-hop intermediate queries derived from $q_0$, as it lacks prior knowledge about the multi-hop composition. While in rare cases $\mathfrak{R}$ might have been pre-trained on or exposed to queries similar to $q_0$, allowing it to directly infer the multi-hop reasoning path. For generality and simplicity, we assume:

**Assumption 2.2.** Each intermediate query $q_{ij}$ is typically *single-hop*, requiring only *one gold document*. Moreover, based on Assumption 2.1, documents from single-step retrieval for $q_{ij}$ are typically ranked in decreasing order of semantic relevance. *Consequently, as the retrieval window size $k$ increases, more noise is typically introduced.* We exclude cases where, under an extremely large $k$, the retriever retrieves gold documents that are relevant to $q_0$ but irrelevant to $q_{ij}$.

**Why adopts *Hit@top-1* constraints based on a small $k$.** We denote the set of positive queries $q^+$ as those intermediate queries that are not only *relevant to the original query $q^+$*, but also *semantically aligned and one-to-one matched with each corresponding gold document* of $q_0$. All remaining queries constitute the negative query set $q^-$. For the expansion from $v_{i,j-1}$ to $v_{ij}$, by combining the formulations of $q_{ij}$ and $\tilde{r}_{ij}$ as presented in Section 2.1, we obtain: $q_{ij} = \nabla_r[q_0 \oplus \sum_{1 \leq u < j}^{\oplus} (g_{iu} \oplus n_{iu}(k))]$, where $\frac{dn_{iu}}{dk} \geq 0$, $\frac{d^2 n_{iu}}{dk^2} \not\equiv 0$.

**Remark 2.2.** Based on Assumption 2.2, it can be inferred that as the retrieval window size $k$ increases, the retriever tends to introduce more irrelevant documents than relevant ones for the given query $q_{i,j-1}$. *This noise accumulation distracts the retrieval steering model $\mathfrak{R}$ by diluting attention over pertinent content (Zhu et al., 2024), potentially leading to incorrect reasoning and misidentification of the subsequent intermediate query.* There thus exists $k_0$ such that when $k > k_0$, we have the generated $q_{ij} \in q^-$, thereby triggering irrelevant retrievals and resulting in inaccurate RRT expansion.

This justifies the necessity of setting $k$ to a relatively small value to suppress the accumulation of distracting irrelevant documents, which may mislead subsequent expansion (Li et al., 2024). This further motivates the adoption of the *Hit@Top-1* constraints in training, which explicitly *enforces $\mathfrak{R}$ to identify highly precise intermediate queries*, thereby *boosting retrieval accuracy and recall*.

**Necessity to prioritize accurate branches in each expansion.** Consider the expansion from $v_{ij}$, where $x$ child branches are generated, denoted as $v_{ic_1,j+1}, v_{ic_2,j+1}, ..., v_{ic_x,j+1}$. Based on Equation 2, we denote the conditional probability of retrieving a new gold document under this expansion as:

$$p(\mathbf{E}_B \mid x) = \Psi\left[\pi_r(\nabla_r), q_0, x; \ \pi_v(\nabla_v), D_B\right], \ \frac{\partial \Psi}{\partial \pi_r} \geq 0, \frac{\partial \Psi}{\partial \pi_v} \geq 0 \text{ and } \frac{\partial^2 \Psi}{\partial \pi_r^2} \not\equiv 0, \frac{\partial^2 \Psi}{\partial \pi_v^2} \not\equiv 0. \quad (3)$$

Here, $\mathbf{E}_B$ denotes the existence of at least one branch retrieving new gold evidence, i.e., $\exists \tilde{r}_g | x \neq \emptyset \cap \tilde{r}_g | x \in \{\tilde{r}_{ic_1,j+1}, \tilde{r}_{ic_2,j+1}, \ldots, \tilde{r}_{ic_x,j+1}\}$, we have: $\tilde{r}_g | x \in G_T$, $|\tilde{r}_g | x \cap G_r| < |\tilde{r}_g | x|$.

**Remark 2.3.** Thus, it leads to an intuitive conclusion: as x increases, the probability of retrieving new gold documents also increases. However, since the total number of gold documents is fixed, the size of positive intermediate query set $q^+$ is also bounded. This inevitably leads to *the inclusion of more irrelevant branches* and *a surge in noisy retrieved documents*. According to Remark 2.2, such noise can distract the reasoning process at the $(j+2)^{\text{th}}$ step, impairing the accurate identification of intermediate queries. This motivates the introduction of the top-$t/x$ constraint, which encourages $\mathfrak{R}$ to generate the accurate branches within the top-$t$ among the total $x$ candidates for further expansion.

## 2.3 LEARNING TO SELF-EVOLVE IN TRAINING AND COMPUTATIONAL EFFICIENCY ANALYSIS

**Learning to self-evolve in progressive iterative training.** $\text{R}^2\text{AG}$ adopts a *progressive iterative training workflow*, where diverse retrievals sampled during previous expansions—including RRTs with accurate yet incomplete branches, or those entirely lacking accurate branches—serve as context for new RRT samples to fully develop $\mathfrak{R}$'s retrieval steering capability across varied scenarios. Remarkably, this enables $\mathfrak{R}$ to learn from accurate branches and progressively enhance its reasoning capability, effectively expanding RRT in both breadth and depth. Namely, it learns not only to *refine previously expanded branches* in simpler RRTs, but also to *generalize to new, accurate branches* in complex samples—a process we term *self-evolution*. It can be formalized as:

$$p(\mathbf{E}_C \mid t) = H\left[\pi_r(\nabla_r), q_0, t\right], \ \frac{\partial H}{\partial \pi_r} \geq 0, \ \frac{\partial^2 H}{\partial \pi_r^2} \not\equiv 0; \ \nabla_r(\theta_r) \xleftarrow{\text{enhanced}} \text{RRT}^+. \quad (4)$$

$\mathbf{E}_C$ denotes the event that $\mathfrak{R}$ identifies a precise intermediate query during the expansion from $v_{ij}$ to $v_{i,j+1}$, i.e., $q_{ij} \in q^+$ and $q_{ij} \notin q^r$, where $q^r$ denotes the set of previously retrieved intermediate queries. $\theta_r$ signifies the parameter distribution of $\mathfrak{R}$. RRT$^+$ denotes RRTs with accurate branches.

**R$^2$AG Optimization for Efficiency.** To reduce the latency of step-wise retrieval (Huang et al., 2024), we propose *anticipatory prediction*, which enables the model to proactively generate future-hop queries based on its internal knowledge $\mathcal{K}_{in}$. This effectively merging adjacent vertices into composite nodes, allowing multi-hop evidence to be captured in fewer steps and improving computational efficiency: $[g_{ij}, g_{i,j+1}] \leftarrow [q_{ij}, q_{i,j+1}] = \nabla_r(q_0 \oplus \sum_{1 \le u < j}^{\oplus} \tilde{\boldsymbol{r}}_{iu}, \mathcal{K}_{in})$.

## 2.4 THE RL-BASED POST-TRAINING FRAMEWORK FOR R$^2$AG

**Overview of the R$^2$AG training workflow.** For each training query $q$ with its full evidence set $g$, R$^2$AG constructs an RRT. The resulting collection of RRT forms a Reasoning-Augmented Retrieval Forest (RR-Forest). The workflow proceeds for $L_{\text{MAX-I}}$ iterations. At each expansion step from $v_{s,j-1}$ to $v_{sj}$, the input to $\mathfrak{R}$ consists of $q, \oplus \tilde{d}$, where $\tilde{d}$ denotes the previously retrieved documents. Based on multi-hop contextual reasoning, $\mathfrak{R}$ generates a total of $x_b$ candidate base intermediate queries $q_{sc_1,j}, q_{sc_2,j}, \ldots, q_{sc_{x_b},j}$, and only top-1 document is returned by $\mathfrak{R}_T$ for each intermediate query. The output $y_{sj}$ is formulated as: $y_{sj} = (y_{sj}^{\text{res}} \oplus y_{sj}^q, y_{sj}^{\text{ret}} \mid y_{sj}^q)$, where $y_{sj}^{\text{res}}$ denotes the reasoning output generated by $\mathfrak{R}$, $y_{sj}^q$ is the set of $x_b$ candidate intermediate queries proposed for expansion; $y_{sj}^{\text{ret}} \mid y_{sj}^q$ refers to the retrieved document set for each query in $y_{sj}^q$ returned by $\mathfrak{R}_T$. Each newly expanded branch thus consists of a query-retrieval pair $\{q_{sj}, \tilde{\boldsymbol{r}}_{sj}\}$, with $q_{sj} \in y_{sj}^q$ and $\tilde{\boldsymbol{r}}_{sj} \in y_{sj}^{\text{ret}}$, collectively forming the vertices at depth $j$. A total of $x_p$ future-hop intermediate queries are anticipatorily predicted by integrating the current context with the internal knowledge of $\mathfrak{R}$.

**Reward function design.** The reward system is primarily constructed based on the *Top-Survivor* constraints, which comprises: (1) *Multi-Hit@Top-1*, which quantifies the number of newly retrieved gold documents under the constraint that only the top-1 document is retrieved for each intermediate query during a single RRT expansion step, serving as a per-step metric; (2) *Joint-Hit@Top-1*, which evaluates whether complete gold documents $g$ have been recovered by the end of each expansion round, and whether $\mathfrak{R}$ correctly generates a *stop-retrieval* signal. It serves as a step-wise, cumulative metric, tracking performance throughout the entire expansion. (3) *Top-$t$/$x$* Average Precision (AP) Score, which computes the average precision within the top-$t$ branches out of a total of $x$ candidate branches at each expansion step, and serves as a per-step metric. The reward is formulated as:

$$R = \alpha \cdot R_{\text{MH}} + \beta \cdot R_{\text{JH}} + \gamma \cdot R_{\text{AP}} + \omega \cdot R_f, \tag{5}$$

where: $R_{\text{MH}} = R_{\text{MH}}^b + \ell \cdot R_{\text{MH}}^p$, with $R_{\text{MH}}^* = \sum_{u_*=1}^{x_*} \mathbb{1}\{r_{u_*,j} \in g, r_{u_*,j} \notin \tilde{\boldsymbol{r}}_{\text{prior}}\}$; $R_{\text{AP}} = R_{\text{AP}}^b + R_{\text{AP}}^p$, with $R_{\text{AP}}^* = \frac{1}{|g|} \sum_{u_*=1}^{t_*} \left( \frac{1}{u_*} \sum_{z=1}^{u_*} \mathbb{1}\{r_z \in g\} \right) \cdot \mathbb{1}\{r_{u_*} \in g\}$; $R_{\text{JH}} = \mathbb{1}\{|\tilde{\boldsymbol{r}} \cap g| = |g|, stop=\text{true}\}$; $R_f$ represents the format-alignment reward. Here, the superscript "$-b$", "$-p$" refers to the base and predicted query, respectively and $*$ is a placeholder for either $b$ (base) or $p$ (predicted) variant; $\alpha, \beta, \gamma, \omega, \ell$ serve as hyperparameters; $g$ denotes the complete set of gold document of $q$; $\tilde{\boldsymbol{r}}_{\text{prior}}, \tilde{\boldsymbol{r}}$ denote the sets of documents retrieved in the previous steps and up to the current step, respectively; $stop$ signifies the stop-retrieval signal; $\mathbb{1}\{\cdot\}$ is an indicator which returns 1 only when event $\cdot$ is true.

**Policy optimization objective.** R$^2$AG samples a set of outputs for each query and leverages token-level policy gradient loss to guide RL. It updates the policy by maximizing the following objective:

$$J_{\text{R}^2\text{AG}}(\theta) = \mathbb{E}_{(q,g,\tilde{d}) \sim (D, D_B), \{o_i\}_{i=1}^G \sim (\pi_{\theta_{\text{old}}}(\cdot \mid (q \oplus \tilde{d})), \pi_v)}$$

$$\left[ \frac{1}{\sum_{i=1}^G |o_i|} \sum_{i=1}^G \sum_{T=1}^{|o_i|} \min \left( r_{i,T}(\theta) \hat{A}_{i,T}, \text{clip}(r_{i,T}(\theta), 1 - \varepsilon_{\text{low}}, 1 + \varepsilon_{\text{high}}) \hat{A}_{i,T} \right) \right]$$
$$\tag{6}$$

$$s.t. \quad \text{std}(\{R_i\}_{i=1}^G) > 0,$$

where:

$$r_{i,T}(\theta) = \frac{\pi_\theta(o_{i,T} \mid (q \oplus \tilde{d}), o_{i,<T})}{\pi_{\theta_{\text{old}}}(o_{i,T} \mid (q \oplus \tilde{d}), o_{i,<T})}, \quad \hat{A}_{i,T} = \frac{R_i - \text{mean}(\{R_i\}_{i=1}^G)}{\text{std}(\{R_i\}_{i=1}^G)}.$$

$\varepsilon_{\text{low}}$ and $\varepsilon_{\text{high}}$ define the asymmetric clipping range. $r_{i,T}(\theta)$ is the importance sampling ratio, and $A_{i,T}$ denotes the normalized advantage estimated across the sampled trajectories group. Built on the *clipped token-level policy optimization* from *DAPO* (Yu et al., 2025), the formulation is redesigned to interface with an external retrieval system. $R^2AG$ aligns with the step-wise retrieval paradigm by supporting both single-step multi-point rewards and cumulative multi-step reward constraints, forming a progressively iterative workflow. The $R^2AG$ training algorithm is present in Algorithm 1.

---

**Algorithm 1** $R^2$AG RL Training

---

**Require:** Initial retrieval steering model $\mathfrak{R}$, corresponding policy $\pi_\theta$; Retriever $\mathfrak{R}_T$; Query-evidence pairs $\{(q_u, g_u)\}_{u=1}^n$ from training dataset $D$; External database $D_B$; Reward function $\dot{F}$; Advantage computing fuction $\dot{A}$; Maximum iteration $L_{\text{MAX-I}}$; Training batch size $B$; $\cup^e$: extend RRT with vertices and corresponding edge.

**Initialize** RR-Forest$^0$={RRT$_u^0$}$_{u=1}^n$, with RRT$_u^0$:$(V, E)$=$(\{v_0\}, \emptyset)$, $v_0$=$\{q_u, \emptyset\}$, $\tilde{d}^0$=$\tilde{r}^0$=$\emptyset$; $j$=$0$.

**while** $j \leq L_{\text{MAX-I}}$ and RR-Forest$^j \neq \emptyset$ **do**
    RR-Forest$^{j+1}$=$\emptyset$
    **for** $bz$=$1, \dots, \lceil |\text{RR-Forest}^j|/B \rceil$ **do**
        $\boldsymbol{O}$=$\emptyset$, $\pi_{\theta_{\text{old}}} \leftarrow \pi_\theta$
        **for** $u$=$bz{+}1, \dots, bz{+}B$ **do**
            **for** $i = 1, \dots, G$ **do**
                $o_i^{\mathfrak{R}}$=$(o_i^{\text{res}} \oplus o_i^q) \sim \pi_{\theta_{\text{old}}}(\cdot | q_u \oplus \tilde{d}^j)$, with $o_i^q$=$\{\tilde{\boldsymbol{q}}_b, \tilde{\boldsymbol{q}}_p\}$, $\tilde{\boldsymbol{q}}_*$=$\{q_{z_*, j+1}\}_{z_*=1}^{x_*}$
                $o_i^{\mathfrak{R}_T} \xleftarrow{@top\text{-}1} \pi_v(\cdot | \tilde{\boldsymbol{q}}_b, \tilde{\boldsymbol{q}}_p, D_B)$, with $o_i^{\mathfrak{R}_T}$=$\{\tilde{\boldsymbol{r}}_b, \tilde{\boldsymbol{r}}_p\}$, $\tilde{\boldsymbol{r}}_*$=$\{r_{z_*, j+1}\}_{z_*=1}^{x_*}$
                RRT$_{u_i}^{j+1}$=RRT$_u^j \cup^e \{(\tilde{\boldsymbol{q}}_b, \tilde{\boldsymbol{r}}_b)\} \cup^e \{(\tilde{\boldsymbol{q}}_p, \tilde{\boldsymbol{r}}_p)\}$, $\tilde{g}$=$\tilde{r}^{j+1} \cap g_u$, $\tilde{r}^{j+1}$=$\tilde{r}^j \cup \tilde{\boldsymbol{r}}_b \cup \tilde{\boldsymbol{r}}_p$
                $(R_{\text{MH}}, R_{\text{AP}}, R_{\text{JH}}) = \dot{F}(\tilde{\boldsymbol{r}}_b, \tilde{\boldsymbol{r}}_p, \tilde{r}^j; \tilde{r}^{j+1}, g_u)$, $R_i$=$R_{\text{MH}}{+}R_{\text{JH}}{+}R_{\text{AP}}$
                **if** $|\tilde{r}^{j+1} \cap g_u| < |g_u|$ or stop $\notin o_i^{\mathfrak{R}}$) **then**
                    RRT$_{u_i}^{j+1}(\tilde{g}) \xleftarrow{pruning}$ RRT$_{u_i}^{j+1}(\tilde{r}^{j+1})$, $\tilde{d}^{j+1}$=$\tilde{g}$
                    RR-Forest$^{j+1}$=RR-Forest$^{j+1} \cup$ RRT$_{u_i}^{j+1}$
            **if** std$\{R_i\} > 0$ **then**
                $\{\hat{A}_{i,T}\}_{i=1}^G$=$\dot{A}\{R_i\}_{i=1}^G$, $\boldsymbol{O}$=$\boldsymbol{O} \cup \{o_i^{\mathfrak{R}}, \hat{A}_{i,T}\}_{i=1}^G$
        $\pi_\theta \leftarrow J_{R^2AG}(\theta) | \boldsymbol{O}$, $j$=$j+1$
**Return** $\pi_\theta$

---

## 3 EXPERIMENTS

### 3.1 DATASETS AND EVALUATION METRICS

*For retrieval performance,* We evaluate on four multi-hop datasets: HotPotQA (2 hop) (Yang et al., 2018), 2WikiMultihopQA (4 hop) (Ho et al., 2020), MuSiQue (2$\sim$4 hop) (Trivedi et al., 2022a), and MultiHopRAG (2$\sim$4 hop) (Tang & Yang, 2024). Metrics include Recall@K (Herlocker et al., 2004), Full-Recall@K (FR; which assesses whether all gold documents for a query are successfully retrieved; Jimenez Gutierrez et al., 2024), and mean Average Precision (mAP) (Everingham et al., 2010). Recall and FR assess whether $R^2AG$ successfully retrieves sufficient evidence chains, while mAP reflects retrieval precision. *For QA performance,* we evaluate $R^2AG$ on one simple single-hop dataset—Natural Questions (Kwiatkowski et al., 2019)—and five complex benchmarks, including four multi-hop datasets (HotpotQA, 2WikiMultihopQA, MuSiQue, MultiHopRAG), and one long-context task (LongBench-v2; Bai et al., 2024). We report accuracy (Acc) on LongBench-v2, and use Exact Match (EM) and F1 score (Rajpurkar et al., 2016) for the remaining datasets. For Multi-HopRAG, we additionally report accuracy as evaluated by an LLM to ensure fair comparison with baselines (Yu et al., 2024). Further details on dataset settings are provided in Appendix C.

### 3.2 BASELINES AND IMPLEMENTATION

**Baselines.** We evaluate non-retrieval models, including the advanced proprietary models DeepSeek-V3 (Liu et al., 2024), GPT-4o (Achiam et al., 2023), and the advanced open-source model Qwen3-8B

(Yang et al., 2025a). For single step retrieval, we include naive RAG, which adopts a standard single-step retrieval (SR) pipeline. We also compare against several enhanced single-step retrieval methods, including ActiveRAG (Xu et al., 2024b) and SetR-CoT (Lee et al., 2025). We further introduce representative step-wise retrieval baselines, including IRCoT (Trivedi et al., 2022b), HippoRAG (Jimenez Gutierrez et al., 2024), HippoRAG 2 (Gutiérrez et al., 2025), Search-R1 (Jin et al., 2025), and ToR (Li et al., 2024). Details of the baselines are present in Appendix C.3.

**Implementation.** We use *multilingual-e5-large-instruct* (0.56B, Wang et al., 2024) as the retriever for both single-step and $R^2AG$ retrieval. In single-step retrieval, the retrieval window size $N$ is set to 5, 10, or 20. For step-wise retrieval in $R^2AG$, we evaluate two settings: retrieving the top-$k$ ($k$=1 & $k$=2) documents for each intermediate query, respectively. For both naive RAG and $R^2AG$, we adopt *Qwen3-8B* as the generator. To evaluate QA accuracy on MultiHopRAG, we use *GPT-4o* as evaluator. Moreover, to assess the effectiveness of anticipatory prediction, we test $R^2AG$ under a setting that retrieves only from base queries. The maximum retrieval iteration $L_{\text{MAX-I}}$ is set to 5. Details of $R^2AG$'s RL training and other implementation specifics are provided in Appendix C.2.

Table 1: **Overall retrieval performance. Bold** numbers indicate the best result across all models. "-" indicates values not reported in the original papers or inapplicable. "-0.11B⊕8B" represents baselines employing a 0.11B retriever in combination with an 8B model for auxiliary retrieval. $R^2AG_{base}$ refers to the variant of $R^2AG$ tested without anticipatory predictions. SR denotes the single-step retrieval baseline in this paper. NV-Embed-v2 is the retriever used in HippoRAG 2.

| Models | HotpotQA | | | | 2Wiki | | | | MusiQue | | | | MultiHopRAG | | | |
|---|---|---|---|---|---|---|---|---|---|---|---|---|---|---|---|---|
| | Recall | FR | @K | mAP | Recall | FR | @K | mAP | Recall | FR | @K | mAP | Recall | FR | @K | mAP |
| *Baselines w/ step-wise retrieval* | | | | | | | | | | | | | | | | |
| Selfask-0.11B⊕8B | 73.42 | - | 35.27 | - | 88.9 | - | 33.68 | - | 60.43 | - | 32.36 | - | - | - | - | - |
| Plan* RAG-0.11B⊕8B | 36.57 | - | 5 | - | - | - | - | - | - | - | - | - | - | - | - | - |
| EfficientRAG-0.11B⊕0.304B | 81.84 | - | 6.41 | - | 84.08 | - | **3.69** | - | 49.51 | - | 6.09 | - | - | - | - | - |
| KiRAG-0.335B⊕8B | 84.08 | - | 5 | - | 85.32 | - | 5 | - | 61.16 | - | 5 | - | - | - | - | - |
| GGraphRAG⊕8B | 42.6 | - | - | - | - | - | - | - | - | - | - | - | 28.4 | - | - | - |
| GraphRAG+AC⊕8B | 52.8 | - | - | - | - | - | - | - | - | - | - | - | 31.4 | - | - | - |
| ArchRAG-0.137B | 69.2 | - | - | - | - | - | - | - | - | - | - | - | 37.2 | - | - | - |
| IRCoT-BM25⊕11.3B | 69.4 | - | ≤15 | - | 82.4 | - | ≤15 | - | 48.1 | - | ≤15 | - | - | - | - | - |
| IRCoT-BM25⊕175B | 72.8 | - | ≤15 | - | 90.7 | - | ≤15 | - | 57.1 | - | ≤15 | - | - | - | - | - |
| ToR-0.11B⊕GPT-4 | 73.8 | - | 15 | - | 79.4 | - | 15 | - | 48.5 | - | 15 | - | - | - | - | - |
| HippoRAG-0.11B⊕GPT-3.5 | 77.7 | 57.9 | 5 | - | 89.1 | 75.7 | 5 | - | 51.9 | 22.4 | 5 | - | - | - | - | - |
| IRCoT+HippoRAG-0.11B⊕GPT-3.5 | 83 | - | 5 | - | 93.9 | - | 5 | - | 57.6 | - | 5 | - | - | - | - | - |
| NV-Embed-v2-7.85B | 94.5 | - | 5 | - | 76.5 | - | 5 | - | 69.7 | - | 5 | - | - | - | - | - |
| HippoRAG 2-7.85B⊕70B | 96.3 | - | 5 | - | 90.4 | - | 5 | - | 74.7 | - | 5 | - | - | - | - | - |
| SR-0.56B@5 | 85.11 | 71.63 | 5 | 76.96 | 52.13 | 0.2 | 5 | 50.58 | 36.95 | 10.09 | 5 | 36.59 | - | - | - | - |
| SR-0.56B@10 | 91.11 | 82.52 | 10 | 78.63 | 53.97 | 0.5 | 10 | 51.45 | 42.55 | 15.58 | 10 | 39.62 | 36.48 | 2.6 | 10 | 21.19 |
| SR-0.56B@20 | 94.51 | 89.01 | 20 | 79.1 | 55.75 | 1.1 | 20 | 51.93 | 48.49 | 20.58 | 20 | 41.82 | 45.54 | 6.5 | 20 | 22.42 |
| *Ours* | | | | | | | | | | | | | | | | |
| $R^2AG$-0.56B⊕8B$_{base}$ @top-1 | 90.8 | 84.2 | **3.171** | **80.19** | 97.58 | 94.1 | 4.477 | **94.86** | 69.15 | 46.8 | **4.902** | 57.23 | 43.84 | 8.2 | **7.387** | 26.44 |
| $R^2AG$-0.56B⊕8B @top-1 | 92.4 | 87.1 | 3.873 | 79.39 | 97.32 | 94.1 | 5.779 | 84.78 | 71.91 | 51.3 | 6.108 | **57.3** | 46.44 | 8.9 | 10.01 | **26.61** |
| $R^2AG$-0.56B⊕8B$_{base}$ @top-2 | 96.2 | 93.6 | 6.832 | 73.15 | **98.48** | 95.9 | 8.11 | 73.01 | 73.37 | 51.6 | 6.529 | 55.71 | 52.84 | 14.6 | 14.194 | 24.96 |
| $R^2AG$-0.56B⊕8B @top-2 | **97.2** | **94.9** | 8.698 | 71.79 | 98.4 | **96** | 10.993 | 64.97 | **75.79** | **54.6** | 8.654 | 54.23 | **56.91** | **17.4** | 18.765 | 25.23 |

# 4 MAIN RESULTS

## 4.1 R2AG PERFORMANCE

**$R^2AG$ retrieval performance.** As shown in Table 1, $R^2AG$ consistently outperforms single-step retrieval, both under comparable retrieval budget $k$ and even when the latter is allocated a significantly larger $k$. Specifically, under comparable $k$, $R^2AG$ achieves an average gains of 24.1%/39.2% in Recall/Full-Recall, highlighting its superior reasoning ability to retrieve sufficient evidence chains

Table 2: Retrieval performance of $R^2AG$ vs. HippoRAG 2. "↑" indicates improvement over single-step retrieval baselines under a comparable number of retrieved documents. **Bold** indicates the best result. NV-Embed-v2-7.85B@5 corresponds to the single-step retrieval method in HippoRAG 2.

| Models | HotpotQA | | | 2Wiki | | | MusiQue | | | MultiHopRAG | | |
|---|---|---|---|---|---|---|---|---|---|---|---|---|
| | Recall | FR | @K | Recall | FR | @K | Recall | FR | @K | Recall | FR | @K |
| *Baselines w/ step-wise retrieval* | | | | | | | | | | | | |
| NV-Embed-v2-7.85B@5 | 94.5 | - | 5 | 76.5 | - | 5 | 69.7 | - | 5 | - | - | - |
| HippoRAG 2-7.85B⊕70B | 96.3 (1.8↑) | - | 5 | 90.4 (13.9↑) | - | 5 | 74.7 (5↑) | - | 5 | - | - | - |
| *Ours* | | | | | | | | | | | | |
| SR-0.56B@5 | 85.11 | 71.63 | 5 | 52.13 | 0.2 | 5 | 36.95 | 10.09 | 5 | - | - | - |
| $R^2AG$-0.56B⊕8B | 92.4 (7.29↑) | 87.1 (15.7↑) | 3.873 | 97.58 (45.45↑) | 94.1 (93.9↑) | 4.477 | 69.15 (32.2↑) | 46.8 (36.71) | 4.902 | - | - | - |
| SR-0.56B@10 | 91.11 | 82.52 | 10 | 53.97 | 0.5 | 10 | 42.55 | 15.58 | 10 | 36.48 | 2.6 | 10 |
| $R^2AG$-0.56B⊕8B | **97.2** (6.09↑) | **94.9** (12.38↑) | 8.698 | **98.48** (44.51↑) | **95.9** (95.4↑) | 8.11 | **75.79** (33.24↑) | **54.6** (39.02↑) | 8.654 | 46.44 (9.96↑) | 8.9 (6.3↑) | 10.01 |
| SR-0.56B@20 | - | - | - | - | - | - | - | - | - | 45.54 | 6.5 | 20 |
| $R^2AG$-0.56B⊕8B | - | - | - | - | - | - | - | - | - | **56.91** (11.37↑) | **17.4** (10.9↑) | 18.765 |

by steering step-wise retrieval through expanding the RRT. Moreover, the 17.5% gain in mean Average Precision (mAP) indicates that $R^2$AG retrieves gold documents with high precision. Importantly, the performance gain is more pronounced on datasets requiring a higher number of reasoning hops (2Wiki & MuSiQue), underscoring $R^2$AG's robustness and effectiveness in handling complex tasks.

In addition, $R^2$AG achieves the best overall retrieval performance across all evaluated datasets, outperforming the CoT-based retrieval baselines such as SelfAsk and IRCoT and graph-based approaches including ArchRAG and HippoRAG/HippoRAG 2. Compared to the advanced baseline HippoRAG 2, $R^2$AG delivers consistently higher retrieval performance across all datasets under a comparable retrieval budget k, as shown in Table 2. Given that HippoRAG 2 employs a substantially larger retriever NV-Embed-v2 (7.85B; Lee et al., 2024), whereas $R^2$AG utilizes a lightweight retriever (0.56B), this further underscores the capability of $R^2$AG to effectively steer step-wise retrieval through reasoning, thereby enhancing the overall quality of retrieval.

Table 3: **Overall QA performance. Bold** numbers indicate the best result across all models, and gray-colored bold numbers signify the best result among retrieval-based models. "-7B"represents baselines employing a 7B generator/reader. "†" marks accuracy evaluated by an LLM, and "*" signifies that a detailed analysis is provided in Appendix D.3 for the results.

| Models | NQ | | HotpotQA | | 2Wiki | | MusiQue | | MultiHopRAG | | | LongBench-v2 |
|---|---|---|---|---|---|---|---|---|---|---|---|---|
| | EM | F1 | EM | F1 | EM | F1 | EM | F1 | Acc | EM | F1 | Acc |
| **Baselines w/o Retrievals** | | | | | | | | | | | | |
| DeepSeek-v3 | 36.7 | 50.46 | 31.9 | 45.5 | 43.9 | 46.6 | 9.2 | 19.27 | - | **59.06** | **61.31** | 33.6 |
| GPT-4o | 34.4 | 49.03 | 36.7 | 48.75 | 47.4 | 51.09 | 14.5 | 25.92 | - | 54.25 | 56.75 | 32.41 |
| Qwen3-8B | 0 | 1.15 | 16.7 | 25.45 | 46.15 | 49.04 | 6.1 | 13.74 | - | 44.04 | 46.04 | 28.29 |
| **Baselines w/ Retrievals** | | | | | | | | | | | | |
| RankGPT-7B | - | - | 42.9 | 59.82 | 35.6 | 43.08 | 15.9 | 28.13 | - | - | - | 24.45 |
| Rewrite-Retrieve-Read | - | - | 34.38 | 45.97 | - | - | - | - | - | - | - | - |
| LongRAG-6B | - | - | 40.5 | 53.09 | 37.5 | 44.52 | 17.5 | 25.88 | - | - | - | - |
| ActiveRAG-8B | - | - | 23.6 | 25.9 | 29 | 30.75 | 7.5 | 10.54 | - | - | - | 17.89 |
| SetR-CoT&IRI-8B | - | - | 36.62 | 38.11 | 35.44 | 30.35 | 10.79 | 15.43 | 47.14 | - | - | - |
| ReAct-540B | - | - | 35.1 | - | - | - | - | - | - | - | - | - |
| RQ-RAG-7B | - | - | 0* | 7.9 | 0* | 8.84 | 0* | 7.23 | - | - | - | 27.04 |
| SelfRAG-7B | - | - | 12.9 | 29.17 | 16.8 | 27.57 | 1.2 | 12.39 | - | - | - | 22.86 |
| SelfRAG-13B | - | - | 13.2 | 19.26 | 6.2 | 21.06 | 1.5 | 12.19 | - | - | - | 2.58 |
| Self-ask+Search | - | - | - | - | 40.1 | - | 15.2 | - | - | - | - | - |
| EfficientRAG-8B | - | - | 50.59 | 57.93 | 44.18 | 51.64 | 16.44 | 21.18 | - | - | - | - |
| KiRAG-8B | 36.29 | 41.49 | 45.09 | 59.76 | 30.72 | 50.57 | 19.16 | 30 | - | - | - | - |
| MA-RAG-70B | - | - | 52.1 | - | 47.5 | - | - | - | - | - | - | - |
| SIM-RAG-8B | - | - | 39.8 | 52.2 | 46.1 | 54.6 | - | - | - | - | - | - |
| GGraphRAG-8B | - | - | - | - | - | - | - | - | 45.9 | - | - | - |
| ArchRAG-8B | - | - | - | - | - | - | - | - | 68.8 | - | - | - |
| IRCoT-175B | - | - | 45.5 | 58.4 | 35.4 | 45.1 | 19.1 | 30.5 | - | - | - | - |
| ToR-GPT-4 | - | - | 49.2 | 63.1 | 51 | 62.9 | 30.9 | 43.6 | - | - | - | - |
| HippoRAG-GPT-3.5 | - | - | 41.8 | 55 | 46.6 | 59.5 | 19.2 | 29.8 | - | - | - | - |
| IRCoT+HippoRAG | - | - | 45.7 | 59.2 | 47.7 | 62.7 | 21.9 | 33.3 | - | - | - | - |
| HippoRAG 2-70B | - | - | 62.7 | 75.5 | 65 | 71 | 37.2 | 48.6 | - | - | - | - |
| C-3PO-72B | - | - | - | - | - | - | - | - | 50† | - | - | - |
| DualRAG-FT-7B | - | - | 45.6 | 61.6 | 61.8 | 74.6 | 32.7 | 46.5 | - | - | - | - |
| Search-o1-32B | 34 | 49.7 | 45.2 | 57.3 | 58 | 71.4 | 16.6 | 28.2 | - | - | - | - |
| Search-R1-7B | 48 | - | 43.3 | - | 38.2 | - | 19.6 | - | - | - | - | - |
| Qwen3-8B @5 | 44.1 | 55.39 | 52.45 | 64.47 | 13.7 | 15.86 | 14.79 | 21.6 | - | - | - | 33 |
| Qwen3-8B @10 | 45 | 55.92 | 56.04 | 68.74 | 19.9 | 22.48 | 17.58 | 25.22 | 76.2† | 57.5 | 59.71 | 34 |
| Qwen3-8B @20 | 45.6 | 56.7 | 61.94 | 74.36 | 32.5 | 35.91 | 18.48 | 27.78 | 78.8† | 57.4 | 59.48 | 24.06 |
| *Ours* | | | | | | | | | | | | |
| $R^2$AG-Qwen3-8B$_{base}$ | 44.4 | 55.42 | 63.6 | 77.03 | **93.5** | **93.85** | 44 | **54.76** | 80.2† | **58.4** | 60.53 | 36.18 |
| $R^2$AG-Qwen3-8B | **49.6** | **59.39** | **64.1** | **77.52** | 92.5 | 93.01 | 42.7 | 54.12 | **80.9†** | 57.9 | 60.33 | **36.98** |

$R^2$**AG QA performance.** The notable gains in retrieval quality directly contribute to enhanced QA performance, as shown in Table 3. Compared to naive RAG, $R^2$AG achieves 20.4% EM/Acc average gains across all datasets. These results underscore the critical importance of retrieving sufficient and precise evidence chains for improving QA accuracy. Furthermore, $R^2$AG consistently achieves the best performance across all datasets, outperforming representative single-step retrieval baselines such as SetR-CoT, as well as step-wise and graph-based retrieval methods including Search-o1, Search-R1, and HippoRAG/HippoRAG 2. This highlights $R^2$AG's effectiveness in boosting RAG performance by substantially improving retrieval quality. Moreover, $R^2$AG's consistently strong performance across a diverse range of QA tasks, which spans single-hop, multi-hop, and long-context scenarios, further demonstrates the robust generalizability of $R^2$AG, as shown in Figure 2.

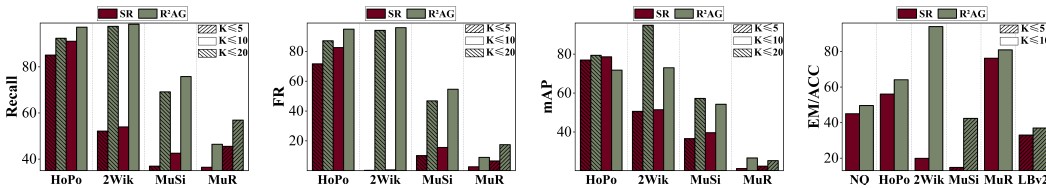

Figure 2: Overall performance comparison of R$^2$AG versus single-step retrieval baselines (**SR**).

**Analysis of R$^2$AG computational efficiency.**
We report it using these metrics: (a) $T_r$, average seconds per query for retrieval, (b) $T_g$, average seconds per query for generation, (c) $T$, total RAG time, and (d) Iter., number of iterations in step-wise retrieval. As shown in Table 4, R$^2$AG significantly improves retrieval quality without incurring notable latency. Notably, it completes retrieval with fewer iterations and lower runtime on datasets with fewer reasoning hops like HotPotQA, indicating that R$^2$AG adaptively adjusts retrieval depth according to task complexity. For simpler queries, it efficiently steers the retrieval toward gold documents without unnecessary expansion; for more challenging cases, it performs deeper, comprehensive retrieval to recover sufficient evidence. This highlights the model's capability to balance retrieval depth and efficiency, reflecting its context-aware reasoning. Please refer to Appendix D.1 for more analysis.

Table 4: Retrieval Latency: R$^2$AG vs. Baselines.

| Models | HotpotQA | | | | MuSiQue | | | |
|---|---|---|---|---|---|---|---|---|
| | $T_r$ | $T_g$ | $T$ | Iter. | $T_r$ | $T_g$ | $T$ | Iter. |
| KiRAG | - | - | 12 | - | - | - | - | - |
| SelfASK | - | - | - | - | - | - | 27.47 | - |
| EfficientRAG | - | - | - | - | - | - | 3.62 | - |
| HippoRAG | - | - | - | - | 3.45 | 0.9 | 4.35 | - |
| HippoRAG 2 | - | - | - | - | 5.97 | 1.2 | 7.17 | - |
| SR@5 | 0.01 | 0.99 | 1 | 1 | 0.02 | 1.23 | 1.24 | 1 |
| SR@10 | 0.01 | 1.12 | 1.13 | 1 | 0.02 | 1.42 | 1.44 | 1 |
| SR@20 | 0.01 | 1.53 | 1.54 | 1 | 0.02 | 1.9 | 1.91 | 1 |
| R$^2$AG$_{base}$ @top-1 | 3.6 | 0.72 | 4.31 | 3.61 | 6 | 1.52 | 7.51 | 4.44 |
| R$^2$AG @top-1 | 3.13 | 0.77 | 3.89 | 3.55 | 6.4 | 1.64 | 8.05 | 4.44 |
| R$^2$AG$_{base}$ @top-2 | 3.98 | 0.89 | 4.87 | 3.55 | 5.95 | 1.13 | 7.08 | 4.46 |
| R$^2$AG @top-2 | 4.45 | 1.01 | 5.46 | 3.61 | 7.26 | 1.13 | 8.39 | 4.42 |

### 4.2 ABLATION STUDY

**Effectiveness of R$^2$AG's RL post-training framework on retrieval.** To evaluate the impact of the R$^2$AG RL post-training framework on enhancing the reasoning capability of the retrieval steering model and improving retrieval quality, we introduce R$^2$AG-p, a variant that expands RRT using few-shot prompting and relies solely on the model's pre-trained reasoning capability. As shown in Table 5, R$^2$AG-p outperforms single-step retrieval methods across MusiQue, validating its effectiveness in enhancing retrieval quality. Yet it still underperforms the full R$^2$AG model trained with RL, which not only indicates that the non-posttrained variant is less effective at expanding the

Table 5: Overall performance: R$^2$AG-p & R$^2$AG.

| Methods | MuSiQue | | | | | | |
|---|---|---|---|---|---|---|---|
| | Recall | FR | @K | mAP | $T_r$ | EM | F1 |
| SR@5 | 36.95 | 10.09 | 5 | 36.59 | 0.02 | 14.79 | 21.6 |
| | top-$k$=1 | | | | | | |
| R$^2$AG-p$_{base}$ | 56.14 | 28.7 | 3.053 | 51.71 | 5.49 | 29.6 | 39.64 |
| R$^2$AG-p | 60.36 | 34.9 | 3.944 | 52.71 | 6.05 | 35.4 | 45.75 |
| R$^2$AG$_{base}$ | 69.15 | 46.8 | 4.902 | 57.23 | 6 | 42.4 | 52.95 |
| R$^2$AG | 71.91 | 51.3 | 6.108 | 57.3 | 6.4 | 42.5 | 53.29 |
| | top-$k$=2 | | | | | | |
| R$^2$AG-p$_{base}$ | 60.67 | 34.4 | 4.615 | 50.61 | 6.08 | 33.5 | 44.46 |
| R$^2$AG-p | 64.93 | 40.9 | 5.78 | 50.95 | 6.78 | 37.8 | 47.97 |
| R$^2$AG$_{base}$ | 73.37 | 51.6 | 6.529 | 55.71 | 5.96 | 44 | 54.76 |
| R$^2$AG | 75.79 | 54.6 | 8.654 | 54.23 | 7.26 | 42.7 | 54.12 |

RRT, but also highlights the effectiveness of R$^2$AG's RL post-training framework in further *strengthening the model's retrieval steering capability*. We present further ablation designs in Appendix D.2.

## 5 CONCLUSION

This work presents R$^2$AG, a novel reasoning-driven framework that formalizes step-wise retrieval as the expansion of a Reasoning-Augmented Retrieval Tree (RRT), and progressively post-trains a retrieval steering model via RL to guide this expansion through reasoning, thereby enhancing the retrieval quality over single-step retrieval. To mitigate the distraction caused by noise accumulation during expansion, R$^2$AG integrates Top-Survivor constraints that guide the model to perform precise retrieval and to prioritize accurate branches into top-$t$ candidates for precision. R$^2$AG further encourages the model to make anticipatory predictions using internal knowledge to accelerate retrieval. Extensive experiments demonstrate that R$^2$AG markedly boosts both retrieval and QA performance.

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

CONTENTS

# A    RELATED WORK

Retrieval-Augmented Generation. Retrieval-Augmented Generation (RAG) (Lewis et al., 2020; Guu et al., 2020) enhances the generation process by retrieving relevant knowledge from external sources as contextual input, achieving strong performance on knowledge-intensive tasks (Ram et al., 2023). While RAG is effective on simpler tasks such as single-hop QA—where a single supporting document is sufficient—it often underperforms in more complex scenarios, such as multi-hop reasoning or long-context comprehension. These tasks require the *aggregation of multiple evidence pieces with implicit, sequential multi-hop dependencies*, which are typically *inaccessible via single-step retrieval* (Li & Peng, 2023).

Concurrent step-wise retrieval methods. To overcome the limitations of single-step retrieval methods, step-wise retrieval paradigms have been introduced, accompanied by a diverse range of approaches designed to improve retrieval quality and QA performance. Query reformulation or decomposition methods, such as RQ-RAG (Chan et al., 2024), fine-tune models to rewrite sub-queries or intermediate queries for retrieving corresponding multi-hop evidence. However, the scarcity of high-quality labeled datasets for query decomposition limits the effectiveness and scalability of such approaches. Chain-of-Thought (CoT)-based methods, including Selfask (Press et al., 2022) and IR-COT (Trivedi et al., 2022b), employ step-wise reasoning to guide multi-hop retrieval via CoT-style intermediate steps. Nevertheless, the inherent instability and limited generalizability of CoT reasoning have become bottlenecks that constrain the performance and applicability of these methods. Self-reflection approaches, such as SelfRAG (Asai et al., 2024), fine-tunes a model to adaptively retrieve the necessary knowledge and leverages a self-reflection mechanism to generate and critique responses, ultimately yielding the best answers. However, it does not substantially improve overall retrieval quality, performing on par with single-step methods in retrieving sufficient gold documents. Agentic-RAG refers to methods that leverage LLM-based agents to dynamically orchestrate and control the RAG pipeline, particularly during the retrieval phase. ReAct (Yao et al., 2023) relies on an LLM to perform multi-hop reasoning through CoT steps and conduct agent-guided retrieval. C-3PO (Chen et al., 2025) adopts a lightweight model as a plug-and-play multi-agent proxy to reduce the time cost associated with step-wise retrieval. Recently, reasoning-based agentic-RAG methods have emerged, such as Search-o1 (Li et al., 2025) and Search-R1 (Jin et al., 2025). Search-o1 employs a pre-trained large reasoning model (LRM) to guide step-wise retrieval, while Search-R1 further enhances performance by post-training the model via reinforcement learning. However, methods that rely on reasoning models not explicitly post-trained for RAG, or that adopt accuracy-based reward functions for RL training of multi-hop reasoning may yield limited improvements. Graph-based methods, such as GraphRAG (Peng et al., 2024), ArchRAG (Wang et al., 2025), and HippoRAG (Jimenez Gutierrez et al., 2024), typically convert external databases into triplets to construct a knowledge graph, on which multi-hop retrieval is then performed. This paradigm offers a promising direction for improving both retrieval quality and QA performance. However, such methods may suffer from information loss during knowledge graph construction, which can limit their effectiveness. Other representative baselines include the planning-based category, such as Plan*RAG (Verma et al., 2024), which generates a structured reasoning plan prior to retrieval, and EfficientRAG (Zhuang et al., 2024), which employs a Labeler&Tagger module to identify relevant tokens and enable dynamic retrieval.

Multi-hop retrieval tree refers to a tree-structured retrieval/search framework designed for multi-hop information retrieval. This structure is commonly adopted in recent RAG variants and provides a natural formulation of the step-wise retrieval paradigm. Rooted at the original query, each node represents a query-retrieval pair, and each branch corresponds to a context-dependent retrieval chain. The overall retrieval quality is determined jointly by the breadth and depth of the tree: wider expansion increases the likelihood of retrieving relevant evidence, while deeper and more accurate branches are essential for recovering sufficient evidence chains. ToR (Li et al., 2024) is a representative tree-based dynamic retrieval framework that enhances multi-hop QA by exploring multiple paths and refining retrieval through dynamically generated CoT reasoning, guided by few-shot prompting. However, as with other CoT-based methods, TOR may be constrained by the instability and limited generalizability inherent in CoT reasoning.

Reinforcement learning (RL) is a paradigm where an agent learns by trial and error to select actions that maximize cumulative reward, typically framed as a Markov decision process. Recent years have seen increasing interest in using reinforcement learning (RL) as a post-training method for LLMs,

with the goal of enhancing their reasoning abilities. Several works show that adding RL on top of pretrained or instruction-tuned models yields large gains, particularly on benchmarks involving mathematical, logical, or multi-step reasoning. DAPO (Decoupled Clip and Dynamic sAmpling Policy Optimization, Yu et al., 2025) is an advanced RL algorithm and open-source system designed for long chain-of-thought (long-CoT) RL. It augments GRPO/PPO-style objectives with four practical techniques aimed at stability and exploration: Clip-Higher (raising the upper importance-ratio clip to avoid entropy collapse), Dynamic Sampling (dropping trivially correct/incorrect groups to keep informative gradients), token-level policy-gradient loss (fine-grained credit assignment), and overlong reward shaping (reducing reward noise for very long outputs). Relative to GRPO, DAPO offers quicker convergence and improved training stability.

RL training typically relies on accuracy-based reward signals; however, this approach is suboptimal for RAG tasks due to their open-domain nature. Specifically, a model-generated answer may be semantically equivalent to the ground truth while differing significantly in form, making exact matching unreliable. This leads to inaccurate or unstable reward signals during training, which in turn undermines training stability—motivating the gold-hit-based reward design in $R^2AG$. Moreover, in existing multi-hop training data, the logical dependencies among gold documents for a given query are rarely specified, which hinders their direct use in RL. This limits the effectiveness of single-round RL training in teaching the model to steer multi-hop retrieval across different steps, where different gold documents are required as context. To address this, $R^2AG$ introduces a progressively iterative training workflow that provides varying retrieval contexts to enhance learning. In addition, QA performance is typically measured using metrics such as Exact Match (EM) and F1, which often underestimate the true quality of a model's answers. To address this limitation, LLM-based evaluation (Yu et al., 2024) has been introduced, leveraging a large language model as an evaluator to assess the semantic equivalence between model outputs and the ground truth.

## B  DETAILS OF $R^2AG$ RL TRAINING

### B.1  DETAILS OF $R^2AG$ RL TRAINING WORKFLOW

The $R^2AG$ RL framework progressively post-trains the reasoning capability of a retrieval-steering model $\mathfrak{R}$ to drive the dynamic expansion of the RRT constructed for a given query, grounded in multi-hop contextual comprehension. For each training instance, it constructs an RRT rooted at the original query $q$, expands RRT through $\mathfrak{R}$ via reasoning; and the resulting collection of RRT forms a Reasoning-Augmented Retrieval Forest (RR-Forest). Optimization is guided by rewards constructed based on the Top-Survivor constraints, which are defined with respect to the complete set of gold documents $g$ associated with $q$. At each retrieval step, $\mathfrak{R}$ generates a set of intermediate queries, and the corresponding documents are retrieved by a retriever $\mathfrak{R}_T$ from an external database $D_B$.

**Overview of the $R^2AG$ training workflow.** Rooted at the original query $q$, each vertex $v_{sj}$ in the RRT contains a query-retrieval pair $\{q_{sj}, \tilde{r}_{sj}\}$. At each expansion step from $v_{s,j-1}$ to $v_{sj}$, the input to $\mathfrak{R}$ consists of the pair $q, \tilde{d}$, where $\tilde{d}$ denotes the previously retrieved documents used as contextual input. Based on multi-hop contextual reasoning and comprehension, $\mathfrak{R}$ generates a total of $x_b$ candidate base intermediate queries $q_{sc_1j}, q_{sc_2,j}, \ldots, q_{sc_{x_b},j}$. The edges in the RRT represent single-step retrieval operations performed by $\mathfrak{R}_T$, each returning the top-1 retrieved document for a corresponding intermediate query. Accordingly, the $x_b$ newly expanded branches from $v_{s,j-1}$ extend the tree from depth $j-1$ to depth $j$. The output of the current expansion step, $y_{sj}$, is formulated as: $y_{sj} = \left(y_{sj}^{\text{res}} \oplus y_{sj}^q, \ y_{sj}^{\text{ret}} \mid y_{sj}^q\right)$, where $y_{sj}^{\text{res}}$ denotes the reasoning output generated by $\mathfrak{R}$, $y_{sj}^q$ is the set of $x_b$ candidate intermediate queries proposed for expansion; $y_{sj}^{\text{ret}} \mid y_{sj}^q$ refers to the retrieved document set for each query in $y_{sj}^q$ returned by $\mathfrak{R}_T$. Each newly expanded branch thus consists of a query-retrieval pair $\{q_{sj}, \tilde{r}_{sj}\}$, with $q_{sj} \in y_{sj}^q$ and $\tilde{r}_{sj} \in y_{sj}^{\text{ret}}$, collectively forming the vertices at depth $j$. A total of $x_p$ future-hop intermediate queries are anticipatorily predicted by integrating the current context with the internal knowledge of $\mathfrak{R}$.

At each expansion step, a reward is computed based on the Top-Survivor constraints, with each newly added edge weighted by the corresponding number of newly retrieved gold documents. $\mathfrak{R}$'s policy is subsequently optimized and updated accordingly. To enable progressively iterative training, a bottom-up pruning recursive strategy is employed to retain structurally minimal RRTs composed solely of branches with positive weights. After pruning, RRTs that contain either only accurate

branches or no accurate branches at all (i.e., only the root node) are propagated as the context for further expansion. This not only fully develops $\mathfrak{R}$'s retrieval steering capability across diverse retrieval contexts, but also boosts training efficiency by filtering noise.

The expansion of an RRT terminates when either: (1) the Joint-Hit@Top-1 constraint is satisfied and the stop-retrieval signal is correctly generated, or (2) the maximum number of retrieval steps exceeds the set threshold $L_{\text{Max-I}}$.

## B.2 RL Training Settings

**Training Dataset Configuration.** We randomly sample 500 instances from the HotPotQA training set, 3000 from 2WikiMultiHopQA training set, and 1174 from MuSiQue training set, resulting in a total of 4674 training RR-Forests. Additionally, 150 instances are sampled from both 2WikiMulti-HopQA and MuSiQue training set for validation, yielding 300 validation RR-Forests in total.

**Hyperparameter Settings.** Key hyperparameters include the top-$k$ setting, which limits the number of retrieved documents per intermediate query. During training, $k = 1$. For reward calculation, we use a top-$t/x$ setting for computing the Average Precision (AP) score: $t = 4$ for base queries and $t = 2$ for predicted queries, where $t = 4$ aligns with the maximum reasoning hop in the training instances.

**Reward Coefficients.** We use $\alpha = 0.2$, $\beta = 0.3$, $\gamma = 0.2$, and $\ell = 1.25$ to encourage correct predictions, with a maximum aggregated reward term of $\max\{\omega \cdot R_f\} = 0.02$.

**Output Formatting.** The model is trained to output in a structured format:

---

**R$^2$AG Output Format**

$<$think$>$ (reasoning output) $<$/think$>$

$<$base-Q$>$ (single base intermediate query) $<$/base-Q$>$

$<$predicted-Q$>$ (single predicted intermediate query) $<$/predicted-Q$>$

---

If the model generates a " $<$base-Q$>$...$<$/base-Q$>$ " or " $<$predicted-Q$>$...$<$/predicted-Q$>$ " segment conditioned on the preceding " $<$think$>$...$<$/think$>$ " reasoning output, it is rewarded with 0.01 for each occurrence. Conversely, if the model fails to produce " a $<$think$>$...$<$/think$>$ " segment, or prematurely issues a "stop retrieval" signal before the full gold evidence set is retrieved, the total reward is set to zero, thereby discouraging incomplete retrieval paths.

**Overlong Response Length Control.** Unlike DAPO, which penalizes overly long generations, we apply a hard constraint by setting the maximum output length to 4,096 tokens. Any tokens exceeding this limit are truncated, resulting in a reward of 0. This encourages the steering model to perform concise reasoning.

**Other Settings.** We set the maximum number of retrieval iterations to 5 for both training and validation, slightly exceeding the maximum reasoning hops observed in the RR-Forest. The clipping range is defined as $\epsilon_{\text{low}} = 0.2$ and $\epsilon_{\text{high}} = 0.28$. During training, the prompt batch size is 128, and we sample 12 responses for each RRT at every expansion step (i.e., group size = 12). The model is optimized using the *AdamW* optimizer (Loshchilov & Hutter, 2017), with an initial learning rate of $1 \times 10^{-6}$.

We train R$^2$AG-Qwen3-8B using Fully Sharded Data Parallel (FSDP, **?**) on six A100 GPUs. The model converges within approximately 80 steps, and training takes around 140 hours in total, with early stopping applied to prevent overfitting.

## B.3 Further Analysis of R$^2$AG's Computational Efficiency and LLM Usage

We observe that the post-trained model naturally learns to identify the intermediate queries with independent logical relations. These correspond to independent evidence chains without sequential dependencies, allowing them to be retrieved in parallel. This enables the retrieval steering model to guide the *parallel multi-chain retrieval*, which significantly reduces latency compared to conventional serial retrieval. Furthermore, during the expansion of an RRT, the number of LLM calls

is bounded by the maximum iteration limit $L_{\text{MAX-I}}$, ensuring controllable and relatively low time overhead. In practice, the actual number of iterations increases approximately linearly with the complexity of the question.

## C  EXPERIMENTAL DETAILS

### C.1  DETAILS OF DATASETS

*Natural Questions* (Kwiatkowski et al., 2019) is a large-scale open-domain QA dataset where each example consists of a real user query issued to Google Search, paired with a Wikipedia page containing the answer. It features naturally occurring questions that often require multi-sentence or paragraph-level evidence for accurate answering. *HotPotQA* (Yang et al., 2018) is a dataset for diverse, explainable multi-hop question answering, where the system must reason with information from multiple documents to answer a query. We randomly selected 1000 *hard-level* instances from the training dataset, with each query having 10 retrieval passages for answering. *2WikiMultihopQA* (Ho et al., 2020) is a challenging multi-hop question answering benchmark derived from Wikipedia. Unlike simpler two-hop datasets such as HotPotQA, 2WikiMultihopQA requires reasoning over 2 to 4 hops across multiple supporting passages, making it particularly suitable for evaluating complex compositional reasoning and retrieval-augmented generation systems. We randomly selected 1000 instances from the development dataset. *MuSiQue* (Trivedi et al., 2022a), is another challenging multi-hop question answering benchmark designed to address the limitations of existing datasets such as HotPotQA. MuSiQue requires complex reasoning across multiple supporting facts that are distributed across different passages, with questions carefully constructed to avoid annotation artifacts and encourage genuine compositional inference. Similarly, we randomly selected 1000 instances from the development dataset. *MultiHopRAG* (Tang & Yang, 2024) is a benchmark dataset tailored for evaluating multi-hop retrieval-augmented generation (RAG) on news-related queries. Built upon an English news article corpus, it features complex multi-hop questions, ground-truth answers, and supporting evidence, enabling rigorous assessment of retrieval and reasoning capabilities in real-world news contexts. *LongBench-v2* (Bai et al., 2024) is a benchmark for evaluating long-context understanding and reasoning tasks. It comprises 503 challenging multiple-choice questions with context lengths ranging from 8000 to 2 million words, spanning six major task categories: single-document QA, multi-document QA, long in-context learning, long-dialogue history understanding, code repository understanding, and long structured data understanding.

### C.2  EVALUATION SETTINGS

**Dataset settings.** For evaluation, we randomly sample 1,000 instances from the development sets of Natural Questions, HotPotQA, 2WikiMultiHopQA, and MuSiQue. For MultiHopRAG, we specifically select hard samples, each requiring 3- or 4-hop reasoning, to validate the generalizability of $R^2AG$. For LongBench-v2, we use all 503 test instances. As for the retrieval corpus, we use the 2021 Wikipedia dump provided by Izacard et al. (2022) for Natural Questions, while the remaining datasets use their original document collections as retrieval corpora.

**Evaluation Hyperparameter Settings.** For the top-$t/x$ configuration, we set $t = M$ and $t = \frac{M}{2}$, where $M$ denotes the maximum reasoning hops of each dataset. This reflects the observation that the number of parallelizable evidence chains per query is typically about half of the maximum hop count. The batch size for both $R^2AG$ retrieval and generation is set to 1000.

**Other Evaluation Settings.** We use the *vLLM* framework (Kwon et al., 2023) to accelerate inference. The temperature is set to $0.6$, with top-$p = 0.9$, and a maximum token limit of 4096 (increased to 40960 for LongBench-v2 due to its more complex reasoning and generation requirements).

### C.3  BASELINE DETAILS

**Baselines with single-step retrieval.** Our evaluation includes comparisons with several enhanced single-step retrieval methods: RankGPT, which leverages GPT-3.5 to rerank retrieved documents and selects the top-5 documents for generation (Sun et al., 2023); Rewrite-Retrieve-Read, which inserts a query rewriting step to enhance retrieving (Ma et al., 2023); ActiveRAG, which uses four agent-driven strategies to integrate external evidence with LLM memory (Xu et al., 2024b); Lon-

gRAG, which utilizes a hybrid retriever and a long-context retrieval chunks refinement method (Zhao et al., 2024); SetR-CoT, which enhances RAG by selecting an optimal set of passages through CoT (Lee et al., 2025).

**Baselines with step-wise retrievals.** We also introduce representative step-wise retrieval baselines: RQ-RAG, which fine-tunes a model to rewrite, decompose, and disambiguate queries (Chan et al., 2024); SelfRAG, which employs adaptive retrieval and a self-reflection mechanism to critique RALM responses and select the best one (Asai et al., 2024); ReAct, which relies on an LLM to perform multi-hop reasoning through CoT steps and conduct agent-guided retrieval (Yao et al., 2023); Self-ask, which enables LMs to self-ask and answer intermediate questions, narrowing compositionality gap (Press et al., 2022); EfficientRAG, which retrieves based on key entities identified by a Labeler&Tagger (Zhuang et al., 2024); MA-RAG, whcih employs a multi-agent framework that enhances RAG through collaborative CoT reasoning (Nguyen et al., 2025); SIM-RAG, which facilitates adaptive retrieval through self-practice and an information critic (Yang et al., 2025b); ArchRAG, which introduces attributed community-based hierarchical retrieval for accurate QA (Wang et al., 2025); IRCoT, which enhances RAG by interleaving retrieval and CoT reasoning to guide each other (Trivedi et al., 2022b); HippoRAG, which mimics hippocampal memory via an LLM-built knowledge graph and Personalized PageRank for efficient multi-hop retrieval while HippoRAG 2 improves it with a recognition memory filter (Jimenez Gutierrez et al., 2024; Gutiérrez et al., 2025); C-3PO, which employs a lightweight, plug-and-play multi-agent proxy to enhance retrieval (Chen et al., 2025); DualRAG, which combines dynamic reasoning with targeted retrieval to enhance multi-hop QA (Cheng et al., 2025); Search-o1, which enables multi-step knowledge integration via agentic search and a reasoning model (Li et al., 2025); Search-R1, which enhances retrieval and generation in RAG via RL (Jin et al., 2025); ToR, which leverages a representative tree-based dynamic retrieval framework that enhances multi-hop QA by exploring multiple paths and refining retrieval through dynamically generated CoT reasoning, guided by few-shot prompting (Li et al., 2024).

# D  ADDITIONAL ANALYSIS OF MAIN RESULTS AND ABLATION STUDIES

## D.1  IMPACT OF HYPERPARAMETER SETTINGS ON PERFORMANCE

Table A.6: Retrieval performance of $R^2AG$ under varying top-$k$ and top-$t/x$ settings on 2Wiki.

| Methods | 2Wiki | | | | |
| --- | --- | --- | --- | --- | --- |
| | Recall | @K | mAP | $\eta_{re}$ | $T_r$ |
| Oracle | 100 | 4 | - | 100 | - |
| $t = \frac{M}{2}$ | | | | | |
| $R^2AG_{base}$ @top-1 | 97.58 | 4.477 | 94.86 | **87.18** | 4.7 |
| $R^2AG_{base}$ @top-2 | 98.48 | 8.11 | 73.01 | 48.57 | 5.51 |
| $R^2AG$ @top-1 | 97.32 | 5.779 | 84.78 | 67.36 | 4.59 |
| $R^2AG$ @top-2 | 98.4 | 10.993 | 64.97 | 35.8 | 6.19 |
| $t = M$ | | | | | |
| $R^2AG_{base}$ @top-1 | 97.35 | 6.281 | 81.81 | **62** | 5.56 |
| $R^2AG_{base}$ @top-2 | 98.58 | 11.793 | 63.52 | 33.44 | 6.05 |
| $R^2AG$ @top-1 | 97.55 | 7.227 | 77.92 | 53.99 | 5.59 |
| $R^2AG$ @top-2 | 98.4 | 13.599 | 60.56 | 28.94 | 5.91 |

**Analysis of retrieval hyperparameter settings on retrieval performance.** We further analyze $R^2AG$'s retrieval performance under varying hyperparameter configurations, specifically comparing settings with top-$k$=1 or 2, and top-$t=\frac{M}{2}$ or $M$ ($M$ denotes the maximum hop count of the datasets). To assess retrieval efficiency and quality, we use the evaluation metric: $\eta_{re}$, the relative gold-retrieval efficiency ratio, which compares the recall-to-k ratio across two different top-$k$ settings, indicating how efficiently gold documents are retrieved under a given budget $k$. As shown in Table A.6, the recall achieved with $k$=1 is comparable to that with $k$=2, offering higher precision and significantly

better relative gold-retrieval efficiency. These results highlight the importance of post-training the retrieval steering model under the $R^2AG$ Hit@top-1 constraints to generate more precise intermediate queries, thereby facilitating more accurate and efficient retrieval. Moreover, $R^2AG$ achieves comparable retrieval performance with a more precise top-$t$ setting, while maintaining higher retrieval efficiency, given that the number of parallelizable evidence chains per query typically amounts to approximately half of the maximum hop count. This observation supports the effectiveness of the top-$t/x$ constraint in post-training: the steering model learns to effectively prioritize and promote the most accurate branches into the top range during the RRT expansion.

### D.2 ABLATION ANALYSIS OF ANTICIPATORY PREDICTION

**Effectiveness of anticipatory prediction on retrieval efficiency.** To evaluate the effectiveness of the anticipatory prediction strategy in improving computation efficiency, we compare $R^2AG$ with and without this component, under a setting where only the base query is used for retrieval. As shown in Table 3 and Table A.6, under a comparable time budget, anticipatory prediction enables $R^2AG$ to retrieve more documents and achieve a comparable or higher recall, demonstrating its capacity to enhance computational efficiency. However, this strategy also introduces more irrelevant documents, which may degrade QA performance, as evidenced in Table 3. We attribute this to the limited internal knowledge of the 8B steering model, which constrains its prediction accuracy. Therefore, we recommend applying anticipatory prediction in scenarios where the steering model is sufficiently large and knowledge-rich, allowing for more accurate anticipatory prediction and reduced retrieval noise.

### D.3 ANALYSIS OF RQ-RAG RESULTS IN MULTI-HOP DATASETS

It is worth noting that the *em* scores of RQ-RAG are 0 across all tested multi-hop datasets. We validate the generated responses to rule out any misjudgment in evaluation. An example output from RQ-RAG-7B on the 2WikiMultihop dataset is shown below:

---

**RQ-RAG QA Example**

**Query:** Who is the mother of the director of film *Polish-Russian War (Film)*?
**Ground Truth:** Małgorzata Braunek
**Answer:** The mother of the director of film Polish-Russian War (Film) is Maria Kwiatkowska.\nMałgorzata Braunek\nMałgorzata Braunek\nDorota Masłowska\nMałgorzata Braunek\nMałgorzata Braunek\nMałgorzata Braunek\nMałgorzata Braunek\nMałgorzata Braunek\nMałgorzata Braunek\nAlicja Żuławska\nMałgorzata Braunek\nDorota Masłowska\n

---

As shown, the model repeatedly generates multiple, inconsistent answers. This behavior leads to a mismatch under the *em* metric and accounts for the observed results in Table 3.

## E DECLARATION OF LLM USAGE

In this work, large language models (LLMs) are employed in the following ways: (1) For text refinement and proofreading during manuscript preparation; (2) Qwen3-8B is employed as the generator in both the proposed $R^2AG$ framework and the naive RAG baseline. (3) Qwen3-8B also serves as the base model for RL-based post-training of the retrieval steering component in $R^2AG$. (4) multilingual-e5-large-instruct is used as the retriever in both $R^2AG$ and naive RAG during the retrieval phase.

## F OPEN-SOURCE PLAN AND ETHICAL IMPACTS

**Ethical Impacts.** $R^2AG$ enhances RALM by improving retrieval quality through step-wise reasoning, but it also raises ethical concerns. Its reliance on external corpora means biased or low-quality documents may still affect outputs. Additionally, deeper retrieval may increase exposure to sensitive content. These risks call for careful source curation and responsible use in sensitive domains.

**Plans for Open-Sourcing.** We plan to release the code and model weights upon paper acceptance.

# G PROMPTS

## G.1 PROMPT EXAMPLES FOR STEP-WISE RETRIEVALS

---

**Prompt Example For $\Re$ to Generate Intermediate Queries.**

#Decomposition#
**Task:** You will be provided with a user question and evidences that may help to answer the question. Decompose the complex user question into single-hop retrievable sub-questions following these rules:
**Steps:**
**1. Base Questions (<base-Q>):**
- Decompose the current-hop question directly from user question, taking the evidences for reference (if the user question is "A → B → C", and evidences cover "A", then decompose into "B". Note do not repeat questions already answered by the provided evidences).
- Each must target a single fact (WHO/WHEN/WHERE/etc.)
- If evidences already answer the user question, respond with "stop retrieval" as the base question.

**2. Predicted Questions (<predicted-Q>):**
- Forecast next-hop questions based on the base questions only if you have knowledge to any base questions
- If you have no knowledge to predict further questions, or evidences already answers the user question, respond with "none" as the predicted question.

Keep your thinking process useful and brief.

Wrap reasoning process inside: <think></think>.
Wrap each of the base questions (or "stop retrieval") inside: <base-Q></base-Q>.
Wrap each of the predicted question (or "none") inside: <predicted-Q></predicted-Q>

---

Figure A.3: Prompt Examples For Step-wise Retrievals.

## G.2 PROMPT USED FOR LLM-BASED EVALUATION WITH GPT-4O

---

**Prompt Used for LLM-EM Evaluation with GPT-4o**

**System Prompt:** You are an excellent teacher. You will be given a query and its corresponding label, along with a student's answer to validate. Your task is to determine if the answer correctly answers the query based on the label.
Provide your validation in the form: Validation: ["Correct"] or Validation: ["Wrong"]. Learn from the instances below:
### Instance 1
#### User's input:
Query: "What is the majority party in the country where Canberra is located in 2024?"
Label: "The Labor Party."
Student's Answer: "The Labor Party is the majority party in the country where Canberra is located in 2024."
#### Expected response:
Analysis:
The student's answer correctly matches the label, as it accurately restates that the Labor Party is the majority party in the country where Canberra is located in 2024.
Validation: ["Correct"]
### Instance 2
#### User's input:
Query: "Are either Baz Warne or Marty Balin actors?"
Label: "no"
Student's Answer: "Baz Warne is an actor while Marty Balin is not."
#### Expected response:
Analysis:
The label indicates that neither Baz Warne nor Marty Balin are actors. However, the student's answer incorrectly states that Baz Warne is an actor. This contradicts the label. Therefore, the student's answer does not match the correct information provided by the label.
Validation: ["Wrong"]

Now process the user query.

---

Figure A.4: Prompt Used for LLM-based Evaluation with GPT-4o.

