# OpenReview forum: "R²AG: Learning to Reason, Retrieve, and Self-Evolve through a Multi-Branch Retrieval Tree"
_ICLR.cc/2026/Conference — Submitted to ICLR 2026_

### Official Review · Reviewer_2m6X · 2025-10-15

**Soundness:** 2
**Presentation:** 2
**Contribution:** 2
**Rating:** 4
**Confidence:** 4

**Summary:**

This paper introduces R²AG, a framework addressing the limitations of standard retrieval-augmented generation (RAG) in handling multi-hop reasoning tasks. The key innovation is formulating retrieval as expansion of a multi-branch Reasoning-Augmented Retrieval Tree (RRT), with expansions steered by a reasoning-driven model post-trained via reinforcement learning (RL). To counter noise accumulation and explosion of irrelevant branches in the expanded tree, a “Top-Survivor” mechanism prioritizes precise branches using gold-hit and average-precision-based constraints. R²AG is further trained in an iterative, self-evolving regime, reinforcing its reasoning strength. Across several multi-hop QA datasets, R²AG demonstrates significant improvements over single- and multi-step retrieval baselines—achieving notable gains in evidence recall, mean average precision, and QA accuracy, while maintaining competitive latency.

**Strengths:**

1. The paper reframes step-wise retrieval as growth of a Reasoning-Augmented Retrieval Tree (RRT) and post-trains a retrieval steering model with RL to expand branches—an explicit, formal treatment that goes beyond prior CoT-style prompting or graph heuristics. The Top-Survivor constraints (Hit@top-1 with AP@top-t/x) and anticipatory prediction are coherent, targeted design choices to curb noise explosion and reduce iterations.
2. The RL workflow is specified with a clear reward decomposition and token-level objective; the training loop interfaces cleanly with an external retriever. Ablations (R2AG-p vs. R2AG) evidence that RL improves over prompted reasoning.
3. Definitions/assumptions (RRT, single-hop intermediate queries, small-k rationale) and the step-wise expansion mechanism are laid out with math and remarks that justify why reasoning should increase gold-hit probability under fixed retrieval settings.
4. Across multi-hop QA datasets, the method boosts recall/FR/mAP and improves downstream EM/Acc while keeping total RAG time competitive; notably, gains are obtained with a lightweight retriever, strengthening practical relevance.

**Weaknesses:**

1. The approach relies on assumptions (e.g., non-duplicate corpora, single-hop intermediate queries, small k to curb noise) that may not hold in open-domain or shifted settings, and the manuscript does not evaluate robustness under near-duplicates, larger k, or BEIR-style domain shifts.
2. The multi-component reward (Multi-Hit@1, Joint-Hit@1, AP@top-t/x, formatting) introduces several weights (\alpha,\beta,\gamma,\omega,\ell), yet the paper lacks sensitivity analyses, seed variance, and failure-mode characterization, limiting conclusions about training stability and generality.
3. Compute reporting is incomplete: latency is provided, but RL post-training cost (GPU hours) and inference FLOPs are not reported; baseline comparisons mix retrievers/generators and retrieval budgets, and equal-budget, shared-retriever settings are missing to isolate the effect of the policy and Top-Survivor.
4. Anticipatory prediction can bias branch growth when future-hop guesses are wrong, and the manuscript does not quantify prediction error rates, downstream impact on FR/mAP/iterations, or safeguards such as confidence-based gating/backtracking.
5. The framework’s capacity to recover when early expansions miss gold documents is uncharacterized; the paper does not measure recovery probabilities or describe diversification mechanisms to escape noisy branches.
6. Ablation attribution is incomplete: while R2AG-p shows that RL improves over prompting, the contributions of Top-Survivor, policy learning, and anticipatory prediction are not disentangled under a shared retriever and fixed retrieval budgets.
7. Empirical scope is QA-centric (plus one long-context setup), so claims of broader applicability to other retrieval-heavy domains are insufficiently substantiated and lack guidance on adapting rewards/constraints.

**Questions:**

See weaknesses.

---

### Official Review · Reviewer_UiW2 · 2025-10-28

**Soundness:** 2
**Presentation:** 1
**Contribution:** 2
**Rating:** 2
**Confidence:** 4

**Summary:**

This paper proposes R2AG (Reasoning-Augmented Retrieval-augmented Generation), a framework designed to enhance retrieval quality in multi-hop question answering by constructing a Reasoning-Augmented Retrieval Tree (RRT). The core idea is to train a retrieval steering model via reinforcement learning (RL) to iteratively expand retrieval branches based on reasoning over previously retrieved context. The authors introduce a Top-Survivor mechanism to select high-quality branches (Hit@Top-1) and a self-evolving training process where the model improves its reasoning ability through progressive RRT construction.

Overall, I find the paper conceptually interesting but difficult to follow due to the unclear definitions, assumptions, and mathematical formulations mentioned in the following. I strongly encourage the authors to clearly and concretely explain these confusing points listed below in the rebuttal to help reviewers better understand the proposed idea and technique. In addition, since the code is not currently provided, releasing it would greatly enhance the paper’s readability and reproducibility by allowing readers to align the described methodology with its actual implementation.

**Strengths:**

- The paper presents an interesting perspective by formalizing multi-hop retrieval as a multi-branch reasoning-augmented retrieval tree (RRT).
- The problem of error accumulation and retrieval noise in step-wise RAG is important and well-motivated.

**Weaknesses:**

1. **Unclear model definition:**

The paper frequently mentions a retrieval steering model, but its identity and training target remain ambiguous. Is this model the retriever (multilingual-e5-large-instruct), the generator (Qwen3-8B), or a joint optimization of both? In the reinforcement learning stage, does the policy gradient optimize the retriever’s embedding space, the generator’s query formulation, or both simultaneously? This distinction is critical for understanding the contribution of the work, and the authors should explicitly clarify in one clear sentence how these components interact.

2. **Confusing assumptions:**



   - **Assumption 2.1** states that the database contains no semantically duplicated documents. This appears redundant, since by definition dense retrieval already maps semantically similar documents to nearby embedding points; as long as no exact duplicates exist, the corpus effectively satisfies this condition. The paper should clarify why this is treated as a formal assumption instead of an implicit property of the dense retrieval setup, and whether it introduces any nontrivial theoretical effect.
   - **Assumption 2.2** is even less clear and, in its current form, appears overly strong and potentially self-contradictory. It seems to imply that each intermediate query can be sufficiently answered by a single “gold” document and that this document will consistently appear at the top of the retriever’s ranked list (i.e., a hit@top-1 assumption). However, this assumption is problematic for two reasons. **First**, in real multi-hop retrieval settings, intermediate queries often require integrating information from multiple partially relevant documents, rather than relying on a single perfect hit. Assuming that one gold document alone can answer the intermediate query oversimplifies the retrieval dynamics and ignores the inherent uncertainty and noise in large corpora. **Second**, if the method indeed relies on the hit@top-1 assumption, it implicitly presumes that both the retriever and the generator are already near-optimal—capable of producing perfect queries and ranking the gold document at the top every time. But if such an assumption held, there would be no need to further optimize the retrieval or reinforcement process, since the system would already perform ideally. Conversely, if the retriever and generator are imperfect (as they always are in practice), then this assumption does not hold, and the entire branching process becomes unstable, as errors will accumulate rapidly along the retrieval trajectory.



3. **Ambiguous mathematical formulation:**

Equations (2)–(4) merely wrap variables within the operator [ ] without explaining what probabilistic distribution or process it represents. There is no indication of how these probabilities are computed—whether they come from the generator’s token distribution, the retriever’s similarity scores, or a policy model’s action probabilities. Furthermore, the partial derivative expressions that follow lack interpretation: their theoretical meaning (e.g., whether they denote policy gradients or score functions) and practical role in the training pipeline are both unclear. As a result, the equations add little explanatory value and make it difficult to connect the theoretical formulation to implementation.

4. **Unclear self-evolution mechanism:**

The paper repeatedly mentions a self-evolve mechanism but does not concretely describe how it operates. Does it involve iterative sampling of negative trajectories, curriculum learning, or experience replay? Is the system generating new data via exploration and re-training, or merely re-weighting existing samples? Without explicit details about what is updated (retriever, generator, or policy) and how, the “self-evolve” term feels descriptive rather than technical.

5. **RL details vague:**


The reinforcement component is not clearly formulated. The reward function (Eq. 5) appears to be a weighted combination of several heuristics—Multi-Hit@Top-1, Joint-Hit@Top-1, and AP-score—but the rationale behind this specific combination is missing. Why should these terms be additive? What relative weights are used, and how sensitive is the method to them? Moreover, Eq. (6) closely resembles GRPO, with the only difference being a modified reward function. The paper should justify in what sense this constitutes an innovation or improvement beyond GRPO, and whether it brings measurable benefits in stability or convergence.




6. **Experimental validity:**

The experiments do not control critical variables such as the retriever, generator, and corpus across baselines. For instance, Table 1 mixes retrieval methods (BM25, NV-Embed, multilingual-E5), making it unclear whether observed gains arise from the proposed algorithm or from stronger backbone models. Fair evaluation requires identical retrieval and generation setups for all methods to isolate the effect of the proposed approach.

7. **Missing or incomplete baseline results:**

Many baselines in Table 1 lack results (“–”) or use values directly cited from prior papers rather than re-evaluated under a consistent environment. This makes it difficult to assess the true improvement brought by R2AG. If some baselines were reproduced locally (e.g., SR-0.56B@k), then their superior single-step retrieval accuracy suggests that rerunning all baselines in the same setup might yield much closer performance, potentially weakening the claimed advantage.

8. **Dataset mismatch:**

The inclusion of LongBench-v2, which primarily evaluates long-context reasoning rather than multi-hop retrieval, seems misaligned with the paper’s objective. If the goal is to test long-context modeling, the comparison should involve established long-context processing methods (e.g., MInference, MemoryBank). Similarly, on the MultiHopRAG dataset, R2AG shows minimal improvement over vanilla RAG, but the paper offers no explanation. The authors should analyze this case—whether it reflects dataset saturation, model limitations, or instability in the branching process.

**Questions:**

1. Could you clearly define what the *retrieval steering model* is? Does the RL optimize the retriever, the generator, or both?
2. What is the motivation behind Assumption 2.1? How is it different from standard dense retrieval assumptions?
3. What exactly does Assumption 2.2 state — that each intermediate query has only one gold doc, or that the gold doc ranks top-1? How realistic is this assumption?
4. Could you mathematically explain how the probabilities in Eqs. (2–4) are computed and what the partial derivatives represent?
5. How is “self-evolve” implemented in practice? Does it involve explicit negative sampling, experience replay, or progressive data expansion?
6. What is the theoretical or empirical motivation for the reward combination in Eq. (5)?
7. In what sense does Eq. (6) differ from GRPO? What are the tangible benefits of your adaptation?
8. Were all baselines re-implemented under the same retriever and generator settings? If not, how can we compare fairly?
9. Why are many baseline results missing?
10. Why is LongBench-v2 used here, and why is R2AG compared to RAG rather than long-context models?
11. Why does R2AG show limited improvement on MultiHopRAG?

---

### Official Review · Reviewer_3j4G · 2025-11-01

**Soundness:** 3
**Presentation:** 2
**Contribution:** 3
**Rating:** 6
**Confidence:** 2

**Summary:**

This paper introduces a novel Retrieval-Augmented Generation (RAG) framework called R²AG (Reasoning-Augmented RAG). The idea is to formalize multi-hop retrieval tasks as the construction and expansion of a Multi-Branch Retrieval Tree (RRT). To overcome the limitations of traditional RAG in complex multi-hop scenarios, R²AG proposes a reinforcement learning (RL)-post-trained retrieval steering model R that generates precise intermediate queries through reasoning, and selects and expands accurate branches based on "Top-Survivor" constraints. Extensive experiments demonstrate that R²AG significantly outperforms existing single-step retrieval methods and several advanced step-wise retrieval baselines on multi-hop retrieval and question answering (QA tasks), achieving substantial gains in recall/accuracy without incurring significant latency.

**Strengths:**

1.	R²AG's formulation of multi-hop retrieval as the construction and expansion of an RRT is intuitive. The integration of RL-post-training, Top-Survivor constraints, and anticipatory prediction creates a comprehensive solution. The construction of the RRT is optimized by training a steering model with RL, where top-survivor method is applied.
2.	The experimental results are compelling, showing substantial performance improvements (24.1%/39.2% gains in Recall/Full-Recall, 17.5% in mAP, and 20.4% in EM/Acc) across various multi-hop retrieval and QA datasets. R²AG consistently outperforms numerous advanced baselines, including CoT-based and graph-based methods.

**Weaknesses:**

1.	The motivations of “top-p survivor” are not covered by the experiments. The paper provides verbal descriptions (Section 2.2 and 2.3) for adopting "Hit@Top-1" (small k) and "top-t/x constraints" (limiting branches x to top-t). However, there is a lack of concrete experimental evidence to demonstrate the specific impact of these constraints on the task performance, or to rigorously validate the sensitivity to k and x values. It's unclear how robust these design choices are without empirical backing.
2.	While anticipatory prediction is mentioned as a mechanism to reduce latency, there is no ablation study to quantify the specific contribution of anticipatory prediction to the reported efficiency gains or its interaction with retrieval quality.
3.	Clarity issues: The paper writing can be confusing. Figure 1 contains some terms that are not mentioned in the main text, like “base branches”. The computation of the reward needs specific demonstrations.
4. It would be nice of the paper to illustrate the depth and the breadth of the RRT. In that case, we can know if the “top-p survivor” prunes the tree or the tree naturally agrees with “top-p survivor”.

**Questions:**

Q1: There are minor notational inconsistencies, such as the use of r~ in the text and formulas for retrieved documents.

---

### Meta-Review · Area_Chair_P2zj · 2026-01-07

**Summary:**

The paper proposes R²AG, a framework that reformulates multi-hop retrieval as the construction of a Reasoning-Augmented Retrieval Tree (RRT), optimized via reinforcement learning (RL) to improve retrieval chains.

Given the severity of the clarity issues and the concerns regarding the fairness of the baseline comparisons, the consensus leans toward rejection.

**Reviewer Concerns:**

No rebuttal has been made, so all the concerns are still outstanding.

**Reviewer Scores:**

Remain unchanged

---

### Decision · Program_Chairs · 2026-01-26

Reject